# Model and Data Concur and Explain the Coexistence of Two Very Distinct Animal Behavioral Types

**DOI:** 10.3390/biology9090241

**Published:** 2020-08-21

**Authors:** Jordi Moya-Laraño, Rubén Rabaneda-Bueno, Emily Morrison, Philip H. Crowley

**Affiliations:** 1Department of Functional and Evolutionary Ecology, Estación Experimental de Zonas Áridas, Consejo Superior de Investigaciones Científicas (EEZA-CSIC), Carrera de Sacramento s/n, 04120 Almería, Spain; rubenrabb@gmail.com; 2Biology Centre of the Czech Academy of Sciences, Institute of Hydrobiology, 370 05 České Budějovice, Czech Republic; 3Department of Biology, University of Kentucky, Lexington, KY 40506-0225, USA; emily.b.morrison@gmail.com (E.M.); pcrowley@uky.edu (P.H.C.)

**Keywords:** frequency-dependent selection, behavioral types, sexual cannibalism, individual-based models, behavioral syndromes, animal personality

## Abstract

Behaviors may enhance fitness in some situations while being detrimental in others. Linked behaviors (behavioral syndromes) may be central to understanding the maintenance of behavioral variability in natural populations. The spillover hypothesis of premating sexual cannibalism by females explains genetically determined female aggression towards both prey and males: growth to a larger size translates into higher fecundity, but at the risk of insufficient sperm acquisition. Here, we use an individual-based model to determine the ecological scenarios under which this spillover strategy is more likely to evolve over a strategy in which females attack approaching males only once the female has previously secured sperm. We found that a classic spillover strategy could never prevail. However, a more realistic early-spillover strategy, in which females become adults earlier in addition to reaching a larger size, could be maintained in some ecological scenarios and even invade a population of females following the other strategy. We also found under some ecological scenarios that both behavioral types coexist through frequency-dependent selection. Additionally, using data from the spider *Lycosa hispanica*, we provide strong support for the prediction that the two strategies may coexist in the wild. Our results clarify how animal personalities evolve and are maintained in nature.

## 1. Introduction

As phenotypic variation is the raw material for evolution [1,2], uncovering the mechanisms behind its maintenance in populations is key to understand the evolutionary process. For instance, ecological resource use may be partitioned among phenotypes in populations, making the population niche to be the sum of smaller individual niches, perhaps contributing to the maintenance of inter-individual phenotypic differences [3]. Phenotypic variation may in turn have important effects in populations, communities and ecosystems [4,5,6], sometimes leading to eco-evolutionary feedbacks [7,8,9]. Although all of the above affects all types of traits, including morphological and physiological, behavioral traits are particularly important because they show some genetic variability, are closely linked to fitness [10] and may be highly labile [2]. These, along with environmental variation across space and time, set the basis for the maintenance of different levels of adaptive plasticity in populations [11]. In particular, low levels of behavioral adaptive plasticity may be explained by the existence of behavioral syndromes [12,13].

Behavioral syndromes [12,13,14,15] emphasize how individual behaviors may be correlated over time or across situations, and that the adaptive value of one type of behavior (i.e., an individual’s set of behavioral traits) may depend on the extent to which this is expressed across contexts. These relationships may have important ecological and evolutionary implications at different levels of organization in animal systems [16] as, for instance, variability in the levels of behavioral plasticity could be constrained by the strength of correlations among traits leading to inter-individual differences in behavior [17,18].

In addition to the cumulative evidence for behavioral syndromes, a few models have attempted to investigate how they evolve and are maintained in natural populations and what selective agents may be responsible for their evolutionary establishment and/or extinction. The maintenance of more than one behavioral strategy has been historically investigated by game theory, which explicitly considers frequency-dependent selection as the mechanism behind the existence of polymorphisms [19,20]. Why are behavioral syndromes pervasive in some populations or species but are replaced by extensive behavioral plasticity in others? In particular, behavioral plasticity (or plastic personalities) may evolve and overcome the decrease in fitness caused by behavioral syndromes, allowing individuals to behave more adaptively in every behavioral context [11]. In addition to models addressing frequency-dependent selection [21,22], other mechanisms have been proposed to explain the maintenance of diverse behavioral types in populations [15,23], including behavioral plasticity [11].

Furthermore, although behavioral syndromes can potentially set the constraints on behavioral plasticity [12,13,15,24,25], in some ecological scenarios, a behavioral syndrome can be good enough, making selection favoring plasticity to be very weak or negligible. This is the case despite the fact that escaping these constraints could be achieved by the evolution of adaptive flexible behavior [26] (see also [27]). Behavioral syndromes can, therefore, maintain the expression of non-optimal behaviors because they are adaptive in a context with a high intensity of selection (e.g., foraging), despite being apparently maladaptive in the other contexts.

A now-classic example of behavioral syndrome is that related to sexual cannibalism; i.e., females killing and consuming males before, during, or after mating [28,29]. This is a behavioral syndrome when genetically-determined female aggressive behavior towards prey is positively correlated with female aggressive behavior towards approaching males. Pre-mating sexual cannibalism could then be a maladaptive by-product of selection favoring fast growth rates and thus voracity in females—the “aggressive spillover hypothesis” (ASH hereafter) [30,31,32,33]. We refer to female voracity sufficiently high to increase her feeding rate but result in attacks on males that may reduce her reproductive success as the SPOV strategy (from SPillOVer). In spiders, for instance, males may be limiting resources as sperm donors for females [30,34], and thus SPOV may seem counterintuitive. However, this behavior may not be maladaptive if the net selective effect of growing to a larger size (and thus producing more eggs) compensates for the risk of remaining unfertilized. Several studies in different spiders already provide support for the ASH and the SPOV strategy, as they show correlations between female aggressiveness and their tendency to attack an approaching male [31,32,33,35,36] (see, however, [37]). In particular, in the fishing spider *Dolomedes triton*, Johnson and Sih [31,32] reported that (1) voracity towards heterospecific prey was correlated with juvenile feeding rate, adult female size, and fecundity; (2) juvenile and adult voracity are positively correlated; (3) voracity towards heterospecific prey is positively correlated with pre-copulatory sexual cannibalism; and (4) individual differences in boldness are maintained and correlated across contexts regardless of predation threat.

Alternatively, sexual cannibalism may occur differently [28,38] if females were able to adjust their attacks on males to female hunger levels [39,40,41,42] or to the availability of stored sperm [43,44,45]. Females that behave according to the latter pattern would follow what we call the “mate first and cannibalize later” strategy (MFCL). Thus, the MFCL strategy is a state-dependent plastic behavioral type [15,23]. There are two ways in which plasticity could occur in the MFCL strategy. Females may be able to discriminate males from other prey and avoid attacking males when in need of sperm. Alternatively, females may tend not to attack prey nor males until they accrue sperm, switching to a relatively high voracity after the first copulation. An important difference between the two strategies is that feeding on males consistently improves female fitness in MFCL [38] (e.g., [43]), whereas adult feeding would not explain differences in female fecundity for SPOV [30]. However, the study by Johnson and Sih [32] suggests that the view of the SPOV strategy could be expanded to include the fitness consequences of female foraging on fecundity.

Across taxa, several studies have already related behavioral types to specific fitness outcomes [46,47,48]. In the case of sexual cannibalism, if we focus on bold and aggressive traits, consistent individual differences across different ecological contexts should be apparent through their effects on the fitness of SPOV and MFCL females. For instance, when acting as prey, due to the impact of predation on population density [34,43,49], SPOV females (i.e., bolder and active foragers) would be expected to suffer higher death rates than the more cautious MFCL females. Moreover, aggressive spiders may be more likely to attack potentially dangerous predators (e.g., other females [50]) further increasing the costs of a SPOV strategy. Hence, this could generate mortality patterns dependent on SPOV and MFCL frequencies, and thus differential selective pressures and trade-offs between personality traits and fitness outcomes in different ecological contexts [32,51]. These context-dependent trade-offs can ultimately be crucial to maintaining genetic variation in behavioral types. In order to take into account all of the above, and understanding the evolution and maintenance of behavioral syndromes, we need a good modeling framework able to consider different ecological scenarios in which to test the fate of different genetically determined behavioral types.

Individual-based models (IBMs) are computer simulations often implemented to address ecological and evolutionary questions [52,53,54]. In these simulations, different individuals and their alleles can be monitored under different ecological scenarios to study the fate of alleles across generations (evolutionary dynamics). Here we identify the ecological scenarios under which the SPOV or MFCL strategies would evolve and be maintained. We do this by building an IBM (Ungoliant 1.0) in which both strategies can potentially invade a population, and we contrast the occurrence and timing of successful invasions in different environments (i.e., high vs. low productivity). To determine whether the strategies predicted by the model to be evolutionarily stable do actually coexist in a natural population, we also analyzed data from field experiments [35,43] on sexual cannibalism in the burrowing wolf spider *Lycosa hispanica* Wackenaer 1837, formerly *L. tarantula* (Linnaeus, 1758), see [55]). We predicted that phenological differences in maturation between the SPOV and MFCL strategies, especially when the former mature earlier allowing greater access to males, would determine which one would prevail and even their potential coexistence. Since SPOV is a behavioral syndrome and MFCL a plastic state-dependent strategy, this work may substantially contribute to our understanding of how behavioral diversity in particular—and phenotypic variation in general—may be maintained in natural populations.

## 2. Material and Methods

### 2.1. The Model

#### 2.1.1. Model Assumptions

Some of the most important assumptions on which the model is based are the following: 1—The population is closed to migration; 2—The population is near carrying capacity and thus there is no population growth across generations. Thus, we set a constant number of adult individuals (N = 1000) each generation and assume that eco-evolutionary dynamics are negligible; 3—Female feeding status does not affect the timing of oviposition, and fitness is not enhanced by early oviposition (but see [43]); 4—The level of aggression is determined by a single sex-linked locus [56]; 5—For implementation, we considered sex determination of spiders to come from a XX (females)–X0 (males) system [57]; 6—Males are pure carriers for the alleles of aggression, and for simplicity, they are all phenotypically identical; 7—There is no male or female choice (but see [24,25]). Previous residency at female burrows determines which male stays and cohabits with the female [58,59]; 8—Females mate only once and are never sperm limited once mated (see, however, [43]); 9—There is no explicit space, and individuals encounter each other randomly; 10—For simplicity, all eggs are of equal size, and thus egg size does not affect offspring quality; 11—As females are territorial [50], competition for prey is negligible; 12—Post-mating sexual cannibalism is negligible, as males almost always escape successfully after mating. Thus, all cannibalistic events occur because females attack approaching males before mating [34,43]; 13—Foraging occurs constantly through time and is only state-dependent in the sense that there is a threshold in energy reserves beyond which females do not feed anymore.

#### 2.1.2. Modeling Algorithm

Figure 1 shows the flow diagram for the model’s dynamics. We simulated hundreds of males and females who encountered each other randomly each day during each season. Depending on strategy (SPOV or MFCL) and (for MFCL only) on condition (mated or unmated), a female will be more or less prone to attack an approaching male. At the end of each season, females that remain alive and who have successfully mated, produce an egg sac that will contribute to the genetic pool of the next generation. If no single strategy takes over the population (i.e., coexistence is maintained), the simulation stops after 20,000 generations. Based on published data, we also set a 1:1 sex ratio at maturation and thus 500 males and 500 females reached maturity each generation [60], as for most solitary spiders in temperate environments [61]. A feeding algorithm allowed each female spider to increase her condition (abdomen width, [62]) each day by a factor dependent on the strategy and on the environment (see Environments below), simulating foraging encounters. The model was written in MATLAB^®^; a copy of the code can be found at http://www.eeza.csic.es/foodweb/Simulators_FWEE.html.

#### 2.1.3. Model Parameterization

*Field parameters of the Iberian tarantula (Lycosa hispanica)*. Parameter magnitudes used in the model came from field data on the Iberian tarantula (*Lycosa hispanica*, formerly *L. tarantula*), a burrowing wolf spider (Table 1). 

*Body condition and food intake.* As the data available for parameterization were morphological, and females were not actually weighed in the field in any of the studies available, body condition and its increments from foraging were incorporated in the model as abdomen width (in mm). Precise estimates of abdomen width would require the density of the nutrients stored in the abdomen as well [65], but this information was unavailable. Since we have the necessary equations relating abdomen width with reproductive output (Table 1), using abdomen width was both simpler and easier.

*Strategies and alleles.* To include the correlational effect of the sexual cannibalism behavioral syndrome across contexts, SPOV individuals (relative to MFCL) obtained more food during the adult stage (higher rapaciousness), experienced higher mortality during both the juvenile and adult stages [12,13,39,49,66], and had a higher propensity to attack approaching males during the mating season (Table 1). The parameter “pspov” determined the probability that a SPOV female attacked an approaching male. Males could then escape with a probability depending on female size (Table 1). Female body size (i.e., carapace width, CW) was determined by its observed relationship to maturation time. To include realistically large SPOV in the simulations, the slope of this relationship was steeper for SPOV than for MFCL. Furthermore, although not explicitly noted in the verbal model of Arnqvist and Henriksson [30], higher voracity and its associated higher growth rate could translate into early maturation, in addition to larger size [67,68]. Thus, we included this feature in the SPOV strategy, and we added a strategy called EARLY-SPOV, in which females matured earlier and to a larger size, for which we shortened maturation time of EARLY-SPOV relative to MFCL by 30%.

In the original Arnqvist and Henriksson verbal model [30], most of the variation in female fecundity was hypothesized to be due to her fixed size at maturation (modern—Araneomorphae—spiders do not molt after reaching the adult instar), and thus sexual cannibalism would have little effect on female fecundity. However, adult feeding in cursorial spiders influences mass relative to fixed size and thus accounts for much of the variation in female fecundity [32,49]. Thus, we also tested Body Condition-Dependent SPOV (BCD-SPOV) and Body Condition-Dependent EARLY-SPOV (BCD-EM-EARLY-SPOV) females, in which their foraging success during their adult life also contributed to their fecundity (Table 1). For simplicity, throughout we refer to all these strategies related to the spillover hypothesis simply as SPOV.

Rather than simulating all juvenile life-stages, we simplified juvenile life by a single parameter “different”, which settled which proportion of SPOV juveniles survived to the adult stage. In principle, the more voracious juveniles (with SPOV phenotype) should experience higher mortality rates and accrue more food [30,32]. However, higher foraging success should allow them to grow at a faster rate, giving them a cannibalistic size advantage (e.g., [43,69]) over small slow-growing MFCL juveniles. Thus, the parameter “different” measures the balance between the cost and the benefit for SPOV juveniles of being more rapacious. This parameter determines which proportion of born SPOV survive relative to MFCL. Since the actual balance may actually depend on predator pressure, which may largely vary across environments, we simulated a large array of “different” values (Table 1).

*Environments*: We considered two distinct environments (poor and rich). A poor environment is characterized by having low food availability, and at least in spiders, a lower density of individuals [70], which in turn will mean a lower rate of encounter between the sexes. Thus, in poor environments, the maximum number of encounters per day for a male (“maxenc”) was low in poor environments (1) and high in rich environments (3). Additionally, in poor environments the daily increase in condition (our proxy of food intake and assimilation) was half of that in rich environments (Table 1). Poor environments also considered higher predation mortality of adult SPOV relative to adult MFCL females, as well as SPOV females attacking approaching males at a higher rate (Table 1).

### 2.2. Simulations

We ran two main sets of simulations. In the first set, we were interested in knowing under which ecological scenarios one or another strategy would be maintained in populations. The second set of simulations was devised to determine the ecological scenarios in which a new mutation determining one strategy could invade a population dominated by the other strategy. The simulations of maintenance helped to identify a meaningful set of parameters that could be used to study the evolution of strategies.

To study the scenarios for the maintenance of the two strategies, we started with a 50% frequency for each of the alleles. We ran all these simulations twice, once considering SPOV (33% *Sm*, 33% *SS* and 33% *mm*) as genetically dominant and another considering MFCL as dominant (33% *sM*, 33% *MM* and 33% *ss*).

We then ran almost identical simulations as the ones before but with SPOV allele frequencies taking values of either 0.01 or 0.99, emulating a novel mutation arising in a population of pure MFCL or in a population of pure EARLY-SPOV respectively. (SPOV vs. MFCL was not investigated because we found that SPOV was never maintained when confronted with MFCL). The aim of these simulations was to reveal the ecological scenarios under which one new mutation could invade the other. Sensitivity analyses, including a wider range of parameters than those in Table 1, allowed us to test the robustness of our conclusions (Appendix A).

#### Field Data: Do the Simulated Strategies Coexist in the Wild?

The data used here come from a 2006 field experiment on the burrowing wolf spider *L. hispanica*, the results of which have been already published [35,43] or are part of a defended PhD [71]. Importantly, these data were gathered after most of the simulations had been run, and therefore the model was not motivated by the data, but vice versa. We here briefly mention the methods to allow an easier interpretation of the results. In a field common garden in which each female received natural prey ad libitum, eighty females were offered a total of 199 males. Half of the females were assigned to a monandry treatment and received males only until they first mated and half of the females were assigned to a polyandry treatment and received three additional males after the female first mated. All males were used only once and offered sequentially (in different days) to the females. We recorded whether the female mated with the male or decided to attack and kill him, or whether neither occurred because the male remained frozen at the female burrow’s mouth. We have previously successfully documented that there is a continuum of female rates of weight gain [35], reflecting a continuum in voracity rates in our common garden, further supported by the positive relationship between the female rate of weight gain (from maturation to the time a female first mated) and the probability that the female attacks the male. This result is consistent with the existence of spiders following a classic SPOV strategy [30] and others following a more docile strategy (low voracity and low propensity to attack males before the female mates) in a natural population of *L. hispanica*. However, since a purely SPOV strategy seems unlikely to be evolutionarily stable in the wild (this paper), we further analyzed the data collected during the above experiment to test whether an EARLY-SPOV strategy (much more likely to be evolutionarily stable in populations) could be present in the studied population. In particular, we tested the hypothesis that females maturing earlier would tend to gain weight at a higher rate in the period between maturation and first mating, which would be indicative of early-maturing females being more voracious. Note that this would not be an artifact of differences in food availability, but a simple consequence of female behavior, as natural prey were equally provided ad libitum to all females in a common-garden experiment [35,43].

In addition, these differences in female behavior were not a consequence of early-maturing females maturing earlier but hungrier, as maturation time was negatively correlated with body condition at maturation (linear model on cubic root body mass and carapace width as covariate [65]; molting date, *b* = −0.0004, *t*_78_ = −2.36, *p* = 0.021; carapace width; *b* = 0.1243, *t*_78_ = 22.77, *p* < 0.0001). This means that early-maturing females were less, not more hungry than late-maturing females. Despite the low number of females that cannibalized males before the female mated (N = 8) [35,43]), we were able to test some additional patterns supporting the hypothesis that two sexually-cannibalistic strategies may coexist in the wild. Data were analyzed using Generalized Linear Mixed Models with female as a random factor when appropriate, and depending on the nature of the data, either Normal or Binomial distributions [72].

## 3. Results

### 3.1. Ecological Scenarios for the Maintenance of Strategies

The MFCL strategy always persisted in populations over the pure SPOV strategy. The SPOV allele went from a frequency of 0.5 to extinction in 4 to 66 generations, depending on the combination of parameters (trajectories not shown). When a Body Condition-Dependent SPOV strategy (feeding affects female fecundity) was contrasted against MFCL, the results were qualitatively the same (Appendix A).

However, when we contrasted a more realistic EARLY-SPOV strategy against the MFCL strategy, we found more variation in which strategy prevailed (Figure 2). First, when the EARLY-SPOV allele was recessive (Figure 2a), it fixated in the population in 4 out of 10 parameter combinations. Fixation of EARLY-SPOV occurred more likely in rich environments (three out of four cases) and when the relative mortality of EARLY-SPOV juveniles was small. As expected, the higher the mortality of juvenile EARLY-SPOV, the lower the probability of persistence for this strategy. When a Body Condition-Dependent EARLY-SPOV strategy (feeding affects female fecundity) was contrasted against MFCL, the results were qualitatively the same (Appendix A).

### 3.2. Frequency-Dependent Equilibria

Interestingly, when the EARLY-SPOV allele was dominant, we found three parameter combinations that resulted in coexistence, with both strategies maintained until generation 20,000- and two quasi-equilibria, in which the MFCL allele disappeared before 19,900. To account for these equilibria, we took the equilibrium that had the greatest oscillation amplitude (“different” = 0.7 for poor environments in Figure 2b, which had a frequency of the EARLY-SPOV allele ranging from 0.58 to 0.98 across the 20,000 generations). We then subdivided the generations according to the frequency of the EARLY-SPOV allele (i.e., BOTTOM: allele frequency < 0.85 and TOP: allele frequency > 0.85), and compared the fitness of individuals following each strategy in the generations when the EARLY-SPOV allele was close to fixation (frequency > 0.85) vs. the fitness of each strategy in all other generations (frequency < 0.85). We performed GLMs with likelihood ratio tests in which each datapoint was one generation. The dependent variable was an estimate of either EARLY-SPOV or MFCL relative fitness. We included the position in the equilibrium (TOP or BOTTOM) as the main factor. Generation number was included as a continuous variable to control for temporal autocorrelation. After controlling for EARLY-SPOV frequency by including the frequency of the EARLY-SPOV allele as a covariate in the model, we found a higher percentage of MFCL females mating at the TOP position of the equilibrium (when EARLY-SPOV was close to fixation: *χ*_1_^2^ = 49.84; *p* < 0.0001; least-squares means ± SE, TOP: 7.67% ± 0.33; BOTTOM, 5.21% ± 0.09). EARLY-SPOV females had lower relative fitness (estimated as the ratio between the mean number of EARLY-SPOV females surviving to maturation and the average between surviving EARLY-SPOV and MFCL females) at the TOP position (1.70 ± 0.01) than at the BOTTOM position (1.81 ± 0.00, *χ*_1_^2^ = 85.4; *p* < 0.0001). These patterns are consistent with coexistence via frequency-dependent selection.

### 3.3. The Evolution of Strategies—The Invasion of Novel Mutations

When EARLY-SPOV was almost the only strategy present in populations (initial allele frequency of 0.99) and its allele was dominant, MFCL could invade in a wide range of ecological scenarios (Figure 3), sometimes leading to coexistence. MFCL females were much more likely to invade in relatively rich environments.

When MFCL was almost the only strategy present in populations (initial allele frequency of 0.99), however, EARLY-SPOV was able to invade only in one scenario: dominance of EARLY-SPOV in rich environments and with the lowest possible juvenile EARLY-SPOV mortality (Figure 4). Invasion occurred rather slowly and after a quasi-equilibrium in which both strategies coexisted for almost 2000 generations.

Detailed sensitivity analyses (Appendix A) showed that the maintenance of EARLY-SPOV depended mostly on the encounter rates between males and females, with this strategy more likely to be maintained when encounter rates were higher. In addition, we found that allowing SPOV and EARLY-SPOV females to use the energy accrued as adults for offspring production (BCD-SPOV and BCD-EM-SPOV) did not change the results qualitatively, as compared with SPOV and EARLY-SPOV, indicating that the timing of maturation rather than fecundity is what confers an advantage to the SPOV strategy. Additionally, relatively low aggression levels towards males in EARLY-SPOV were less likely to allow this strategy to prevail in populations. Finally, increasing prey availability in a poor environment also allowed MFCL to exclude EARLY-SPOV when the juvenile mortality of the latter was lowest and the EARLY-SPOV allele dominant.

#### Field Data: Do the Simulated Strategies Coexist in the Wild?

We found that, prior to mating, there was a negative correlation between the time to maturation (independent variable) and the rate of weight gain in the period after maturing and before mating (GLM, *b* = −888.9, *t*_78_ = −5.71, *p* < 0.0001; Figure 5). This reflects differences in voracity between early- and late-maturing females, consistent with the existence of EARLY-SPOV and MFCL females in the population. Further, by dividing females into those that gained weight and those that lost weight before mating, we found that the ones gaining weight after maturing but before mating had matured 20 days earlier than those that lost weight during the same life stage (GLM, likelihood ratio test, χ_1_^2^ = 25.8, *p* < 0.0001, Figure 6), a figure that is very close to our simulated maturation differences between EARLY-SPOV and MFCL females of 17 days. However, the spiders that lost weight before mating compensated for their relative losses by switching to a much higher rate of food acquisition after mating, relative to those that gained weight at a high rate prior to mating (GLM on the difference between the rate of weight gain after first mating minus the rate of weight gain before first mating, likelihood ratio test, χ_1_^2^ = 82.2, *p* < 0.0001, Figure 7). In light of these results, we propose that both the magnitude of switching and the timing of maturation can indicate whether a female is closer to an MFCL or to an EARLY-SPOV strategist. Early-maturing females with low switching rates (i.e., indicating that they cannot modulate their foraging rate across contexts, before vs. after first mating) would be closer to EARLY-SPOV, whereas late-maturing females with high switching rates would be closer to MFCL females. Note, however, that this type of MFCL females would differ from the MFCL females that have been modeled in our simulations, as in our field data, due to their flexible behavior, MFCL females would be able to compensate their relatively low feeding rates before mating by switching to higher feeding rates after mating (i.e., adaptive plasticity). Further compensation for the former relatively low rate of food acquisition may be achieved by feeding on any additional male that approaches a female after she has mated [43]. To further confirm the above we classified the females that were experimentally offered additional males after mating as early cannibals if they had attacked a male before first mating (EARLY-SPOV), as late cannibals if they had attacked a male after first mating with another (MFCL) and as neutral if they had never attacked a male. We predicted that, if early cannibals were consistent with an EARLY-SPOV strategy, they should switch their rates of weight gain between contexts (before vs. after mating) less than late cannibals. The latter, to be consistent in turn with a behaviorally-plastic MFCL strategy, could compensate previous mass losses during the period in which they were waiting to mate by switching to a higher rate of weight gain after first mating. Indeed, while early cannibals had a switching value near zero (i.e., they maintain an equal rate of weight gain before and after mating), late cannibals had a significantly higher switch value than early cannibals (GLM, χ_1_^2^ = 5.1, *p* = 0.024, Figure 8). Hence, after first mating, switching to a higher rate of attacks towards males correlated with a higher rate of weight gain (not explained by feeding on the attacked males as they were removed immediately from their jaws [35,43]) and therefore of attacks on heterospecific prey, while females that cannibalized males before mating maintained a constant rate of weight gain (voracity) across contexts, consistent respectively with the coexistence of MFCL and EARLY-SPOV strategies in the population. Interestingly, 6 out of the 17 females fitting the MFCL strategy (those that killed a male only after mating) and zero out of the seven females fitting the EARLY-SPOV strategy lost weight before mating (Likelihood-ratio test, χ_1_^2^ = 4.92, *p* = 0.027), suggesting that MFCL females may even stop feeding while waiting to mate with a male. Therefore, MFCL females could potentially be using adaptive plasticity, stopping feeding until a suitable male arrives and then immediately switching to be highly voracious right after mating. If that were the case, we could further detect the presence of MFCL females, if those females that switch to a high voracity after mating are more likely to kill subsequent approaching males. These would be the females that are docile and mate first, but once mated turn voracious and cannibalize subsequent approaching males. This relationship was found in our data (binomial GLMM with female as random factor, χ_1_^2^ = 4.88, *p* = 0.027, Figure 9). To further confirm that the degree of switching may be an indication of the female strategy, females switching to higher feeding rates after first mating were the ones that also matured later, consistent with the MFCL strategy (GLM, χ_1_^2^ = 8.6, *p* = 0.003, Figure 10).

## 4. Discussion

Our results provide compelling evidence for how a state-dependent plastic strategy and a behavioral syndrome, two very distinct behavioral types, may coexist in the wild via frequency-dependent selection. These add up to other models and mechanisms explaining the evolution and maintenance of behavioral types in populations (e.g., [21,22,23,73]). The difference in our study is that we parameterized a model with data from a real system, and further data collected in that same system provided evidence for the predictions of the model. Therefore, sometimes rather than models of great generality, the particularities of the life history of a species used to build models à la carte can be useful to explain the coexistence of very distinct behaviors. This feedback research between real systems and IBMs entails a “feedback research program” [7,74]. Our results are an example of such program in which we went from data to a model and back to data. The data collected even suggested ways for further tuning the model, as the field data showed how MFCL females have mechanisms to compensate for the energy lost while waiting for mates.

We found that in natural populations, an adaptive strategy in which a female does not kill and/or consume males until she has obtained sperm [28,38,43]—our Mate First and Cannibalize Later strategy-will prevail over a pure spillover strategy [30,31,32]. However, a spillover strategy in which higher levels of attacking heterospecific prey lead to higher growth rates and larger, earlier-maturing adults resulted in a wide range of scenarios in which this EARLY-SPOV strategy could be maintained in natural populations instead of the MFCL strategy. We even found ecological scenarios in which this EARLY-SPOV strategy could invade the previously prevailing MFCL strategy over evolutionary time. However, the MFCL strategy was able to prevail and to invade the EARLY-SPOV strategy in the majority of ecological scenarios. In addition, we found that, under some circumstances, there was a frequency-dependent outcome in which both strategies can coexist in nature, and we investigated whether indications of coexistence of these two strategies could be found in a natural population.

Importantly, the analysis of empirical studies of *L. hispanica* [35,43] showed that these two strategies can coexist in a wild population. In accord with the model, we found (1) a negative correlation between the time of maturation and an estimate of voracity (the rate of weight gain [35]), and that (2) the females that gained weight before mating (i.e., they were actively foraging) matured 20 days earlier than the females that lost weight (very close to the 17 days difference between the simulated strategies). Additionally, we found that the presumed MFCL females (i.e., those that had killed a male only after mating) used adaptive foraging plasticity, maturing later and mating with a male before switching to become highly voracious towards both males and heterospecific prey.

The sensitivity analysis of our simulations showed that the key advantage for the EARLY-SPOV strategy was having a low differential predation rate upon spillover juveniles (the parameter “different”) and thus a low cost for boldness. Additionally, the high encounter rate between males and females results in reduced male densities for the late-maturing MFCL females, because EARLY-SPOV females have higher chances to mate and kill males early in the season. Data on the encounter rate of males and females and the rate of sexual cannibalism in sexually cannibalistic species support this idea. The crab spider *Misumena vatia* has a low rate of premating sexual cannibalism (<7.6%, [75]), in accord with low male-female encounter frequencies (<1 female every two days [76]). Thus, for *M. vatia*, “maxenc” = 0.5. However, sexual size dimorphism is extreme in this species, with females being several times larger than males [77], perhaps making sexual cannibalism less nutritionally valuable for females [78]. However, in fishing spiders of the genus *Dolomedes* [30,31,32], the encounter rate must be much higher. This is because females do not defend central territories, but raft around on ponds moving rapidly [79]. Additionally, higher encounter rates may be a by-product of foraging movements increased to meet the energy demands of egg production [80]. Since, with few exceptions [81], spider males are the searching sex and move around at a higher rate than females [82], the rate of encounter between the sexes must be very high, making the spillover strategy more viable in these spiders. As we gather more information on the natural rate of encounter between the sexes across sexually cannibalistic species, we will be better able to test the hypothesis that a higher encounter rate leads to higher female aggression levels.

The aggression levels of spillover females need to be high for this strategy to be maintained in populations. The sensitivity analysis showed that reducing the probability for an EARLY-SPOV female to attack a male resulted in two new scenarios in which MFCL persisted. This makes sense, as low aggression means low male mortality and no shortage of males for MFCL females. However, our results also showed that behaving less aggressively towards males in poor environments can benefit EARLY-SPOV, likely because scarce prey more strongly affects MFCL females.

The availability of alternative prey also helped determine which strategy prevailed, benefiting MFCL females only in one additional ecological scenario. Furthermore, including Body Condition-Dependent SPOV or BCD EARLY-SPOV females, in which adult feeding affected offspring production, produced results that were not qualitatively different from the basic scheme (see [30]). This result is counterintuitive, as sexual cannibalism may be a strategy to alleviate food limitation [39,43,83], and adult foraging has been shown to greatly affect spider fecundity [43,49], even in a fishing spider [32]. Additional simulations including variation in the quality of the prey could help to solve this apparent paradox.

A potential drawback of our simulations is that the levels of aggression and boldness were determined by a single sex-linked gene [56], while behavioral syndromes are quantitative in nature. However, since we performed a detailed sensitivity analysis in which we explored the levels of correlation among behaviors and ecological constraints (e.g., probability of attacking males, foraging success, differences in mortality between strategies), we think that including a more complex quantitative genetic basis for the two strategies (e.g., [7,74]) would have resulted in qualitatively similar results.

### 4.1. Ecological Determinants of the Frequency-Dependent Equilibrium

Despite the paucity of empirical evidence (e.g., [84]), negative frequency-dependent selection is thought to be important in the maintenance of behavioral polymorphisms in natural populations [23,85,86,87,88]. Negative frequency dependence may sustain mixtures of different behavioral phenotypes and allow populations to evolve towards equilibria in which different strategies have similar expected fitness functions [19].

Therefore, understanding which ecological factors favor the evolution of these equilibria may be of central importance. A close study of the conditions associated with coexistence in our results allowed us to identify the main ecological mechanisms (Appendix A). First, male survival was low (more males died and in a shorter time) at high frequencies of highly aggressive EARLY-SPOV females (TOP position in the equilibrium), as the earliest maturing females within the EARLY-SPOV strategy had more access to males than the latest maturing females. Furthermore, MFCL females, by having a less-aggressive strategy, benefitted when males were scarce at high EARLY-SPOV frequencies. This allowed MFCL females to enjoy relatively higher mating success and thus higher fitness at the TOP position of the equilibrium. It is remarkable that small differences in fitness and mating success can produce a frequency-dependent equilibrium, suggesting that demonstrating the causes of coexistence in nature may turn to be very challenging. A low frequency of spillover females and differences among populations may explain the failure to find evidence for the ASH in some studies (e.g., [37]).

### 4.2. Behavioral Syndromes and Behavioral Plasticity

The existence of behavioral syndromes points to limits of adaptive behavioral plasticity, as the correlation of behaviors across contexts implies that behavior is not optimized in all contexts [12,13]. The spillover strategy exemplifies a behavioral syndrome in which selection for high levels of aggression leading to fast growth in juveniles and large adult body sizes (and thus higher fecundity) may lead to a decrease in fitness in the mating context. This is especially true when males are a scarce resource and are killed by these highly aggressive females even when these females have not yet obtained sperm [12,13,30,31,32]. However, we show how female spiders may express behavioral plasticity by lowering their foraging rate even to the point of losing weight, and that this strategy (MFCL) may coexist with (early) spillover individuals. Indeed, Neff and Sherman [89] suggest that animals can be behaviorally plastic in many situations in which constraints can be neglected, with the behavior that is most rewarding in terms of fitness displayed in each context. However, there are surely limits to behavioral plasticity (e.g., [90,91,92]), and this may explain why behavioral syndromes have in fact been documented (e.g., [93,94,95]).

Until now, researchers thought that in spiders, females could achieve behavioral plasticity in the context of sexual cannibalism through either (1) distinguishing males from other potential prey, or (2) decreasing general aggression levels until they mate at least once [38,96]. If mate recognition is constrained, and females cannot distinguish conspecific males from heterospecific prey [96], then females can only be plastic by decreasing aggression levels against both males and prey. Decreasing the rate of attack upon heterospecific prey can be highly costly, because female spiders are often food limited [70]. Interestingly, in *Dolomedes triton,* there is a time window of a few days during which recently matured females show very low levels of aggression, switching back to high aggression levels afterward regardless of whether they have mated or not (Nancy Kreiter, *pers. comm.*), suggesting limitations in mate recognition and thus limits to plasticity. Here, we reported how the presumed MFCL females in a wild population of *L. hispanica* are likely to decrease their voracity even to the point of losing weight. However, we also show how the plasticity of this strategy allows these females to adaptively switch to a very high rate of food acquisition (both of males and heterospecific prey) once they have mated, partially compensating for their previous losses. This behavioral plasticity was not shown in early-maturing *L. hispanica* females, matching the spillover genotype.

## 5. Conclusions

We identified ecological scenarios that can maintain a version of the spillover behavioral syndrome in which spillover females not only grow to a larger size but also mature earlier. The main ecological determinants favoring this early-spillover strategy are low predation rates on both offspring and adults and a high encounter rate between males and females. However, in most ecological scenarios, an alternative strategy in which females wait to have sperm to attack males or attack them only if food availability is low, seems to prevail. Consistent with the finding that the two strategies can coexist over the long term, we found evidence that these strategies are present in a population of the burrowing wolf spider *L. hispanica*. Our results increase our understanding of how behavioral syndromes can be maintained in natural populations, as well as what determines the maintenance of behavioral types (personalities). We also detected the ecological scenarios that allow the coexistence of two rather extreme strategies via frequency-dependent selection, for which there are few examples in nature.

## Figures and Tables

**Figure 1 biology-09-00241-f001:**
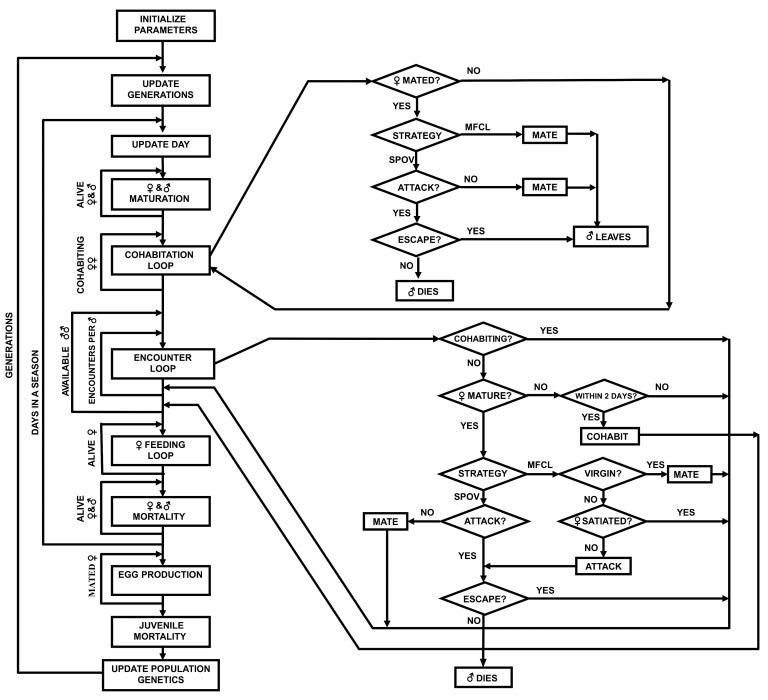
Model flow diagram. Each day (see “ENCOUNTER LOOP” starting in the center of the diagram and flowing to the right) a male randomly encounters females whose genotypes determine their cannibalistic strategy (Mate First and Cannibalize Later—MFCL or SPillOVer—SPOV). SPOV females attack the males regardless of whether females have previously mated or have reached satiation. MFCL females attack males only if these females have previously mated or if they are not satiated. Males have some probability of escaping a female attack. Males may expend a few days cohabiting (top right loop) with virgin females [34,63], and each day males have a probability of mating or being attacked by this female, which will also depend upon the female strategy. Each day females feed on alternative prey, depending on strategy (SPOV feeding at a higher rate than MFCL). Associated mortality rates for males and females depend upon strategy (SPOV higher mortality rates than MFCL). At the end of the mating season, females reproduce according to their strategy, with offspring mortality also contingent on the strategy (SPOV higher mortality than MFCL). See text and Table 1 for further details and model parameterization.

**Figure 2 biology-09-00241-f002:**
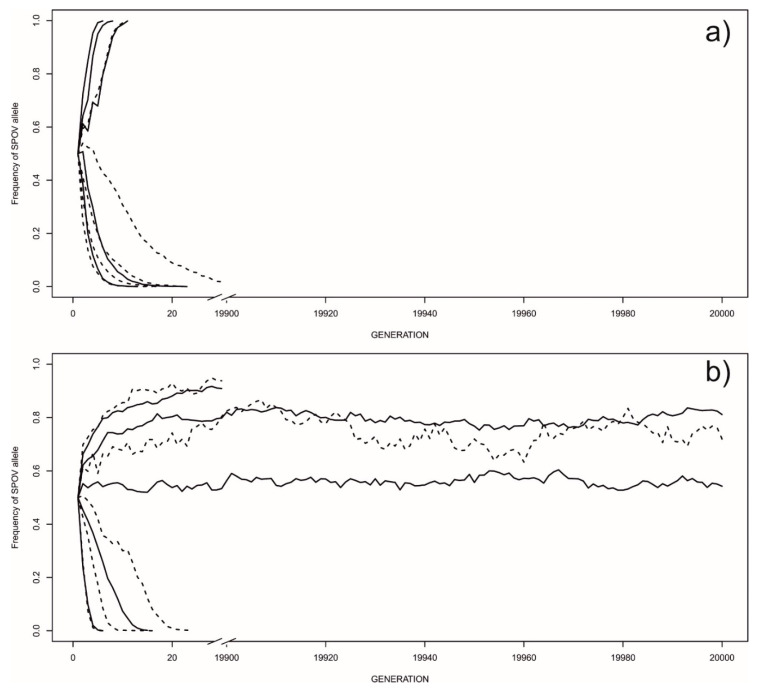
Maintenance of either the EARLY-SPOV or the MFCL strategies depending on the type of environment. The *Y*-axis shows the frequency of the EARLY-SPOV allele (*p*) relative to the MFCL allele (*q* = 1 − *p*). Thus, zero frequency for EARLY-SPOV means 100% presence of MFCL. From bottom to top, and within each habitat type (rich or poor, respectively depicted by solid and dashed lines), the parameter “different” takes the values 0.1, 0.3, 0.5, 0.7 and 0.9 (A value of 0.5 means that for every 100 MFCL juveniles that survive until maturation, only 50 EARLY-SPOV survive). (**a**) Allele dynamics when EARLY-SPOV is recessive; (**b**) Allele dynamics when EARLY-SPOV is dominant (the arrows indicate points of extinction for the MFCL allele). Note the frequency-dependent equilibria that allow both alleles to persist in populations for thousands of generations. The two top curves interrupted at the break are two quasi-equilibria in which the MFCL strategy was eventually driven to extinction before generation 19,900.

**Figure 3 biology-09-00241-f003:**
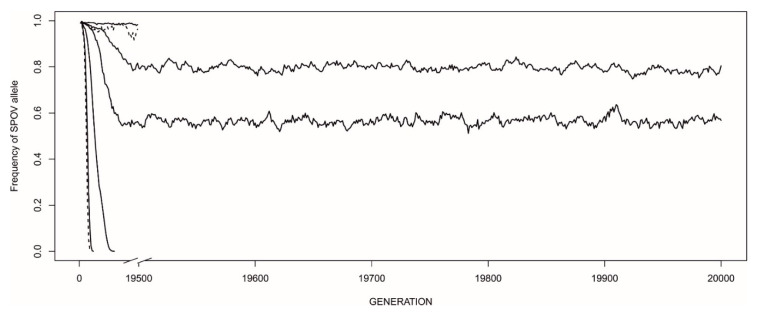
Patterns of invasion of the MFCL strategy in a population in which 99% of individuals are EARLY-SPOV. The *Y*-axis shows the frequency of the EARLY-SPOV allele (*p*) relative to the MFCL allele (*q* = 1 − *p*). Thus, zero frequency for EARLY-SPOV means 100% presence of MFCL. From bottom to top, and within each habitat type (rich or poor, respectively depicted by solid and dashed lines), the parameter “different” takes the values 0.1, 0.3, 0.5, 0.7 and 0.9 (see Figure 2 for interpretation). The two top curves interrupted at the break are two quasi-equilibria in which the MFCL strategy was eventually driven to extinction before generation 19,500.

**Figure 4 biology-09-00241-f004:**
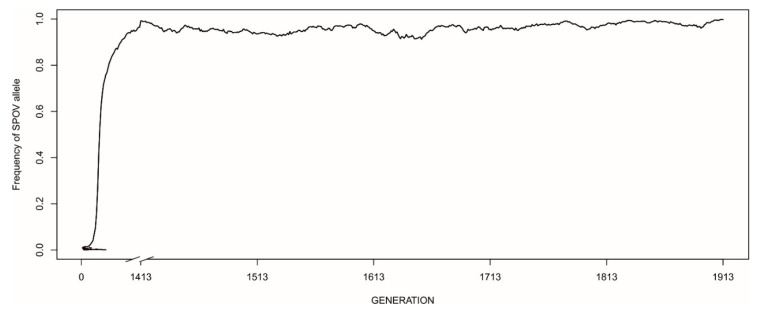
Patterns of invasion of the EARLY-SPOV strategy in a population in which 99% of individuals are MFCL. The *Y*-axis shows the frequency of the EARLY-SPOV allele (*p*) relative to the MFCL allele (*q* = 1 − *p*). Thus, zero frequency for EARLY-SPOV means 100% presence of MFCL. The MFCL strategy prevails in all scenarios but one: rich environment with parameter “different” taking the value 0.9 (meaning that 90 out of each 100 born juveniles of EARLY-SPOV survive until maturation).

**Figure 5 biology-09-00241-f005:**
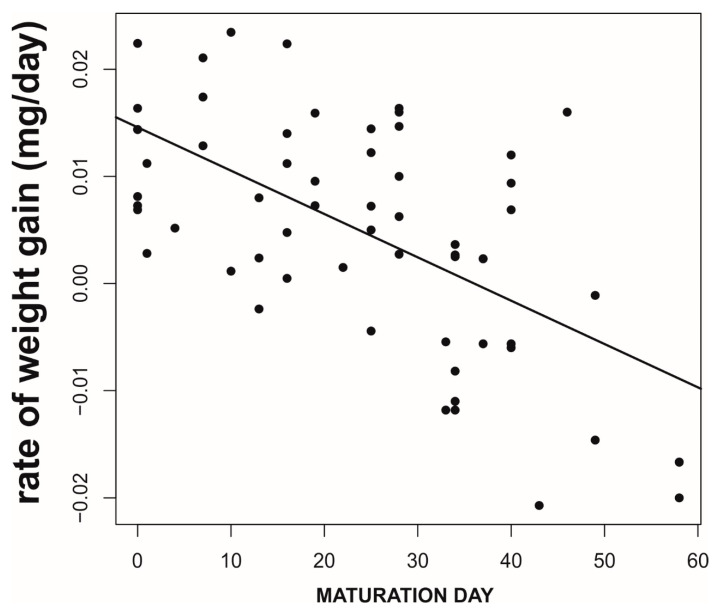
Negative relationship between female age at maturation and the rate of weight gain after maturation and before mating. Data correspond to the burrowing wolf spider *L. hispanica* [35,43].

**Figure 6 biology-09-00241-f006:**
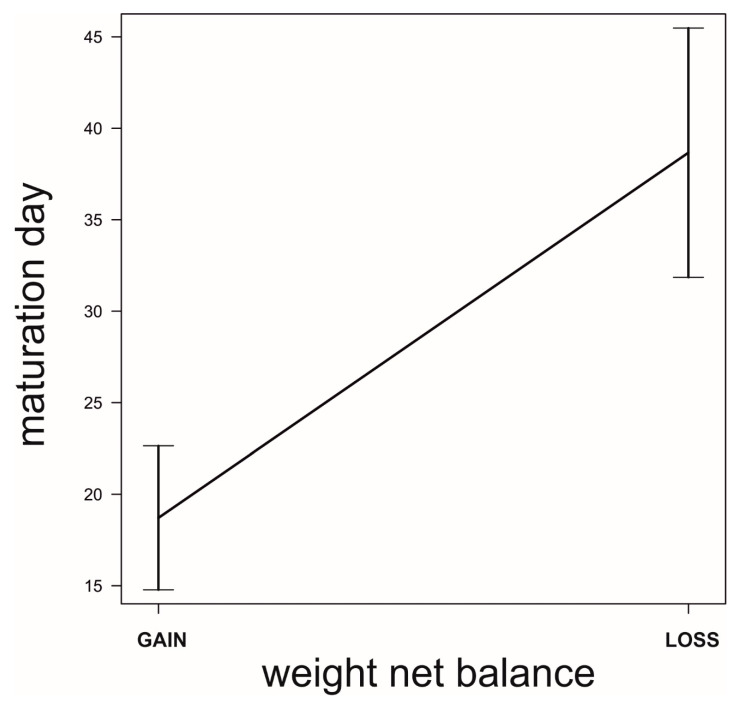
Differences in molting dates between females that lost weight before their first mating and those that gained weight (and were clearly actively foraging) before their first mating took place. Data source as in Figure 5.

**Figure 7 biology-09-00241-f007:**
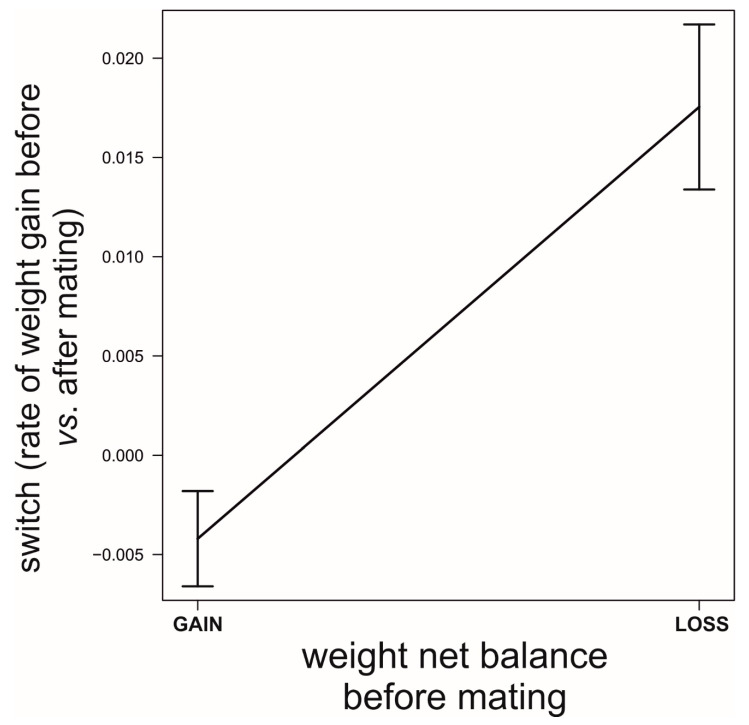
Differences in switching in the rate of food acquisition (difference before vs. after first mating) between females that lost weight before their first mating and those that gained weight (and were clearly actively foraging) before their first mating took place. Data source as in Figure 5.

**Figure 8 biology-09-00241-f008:**
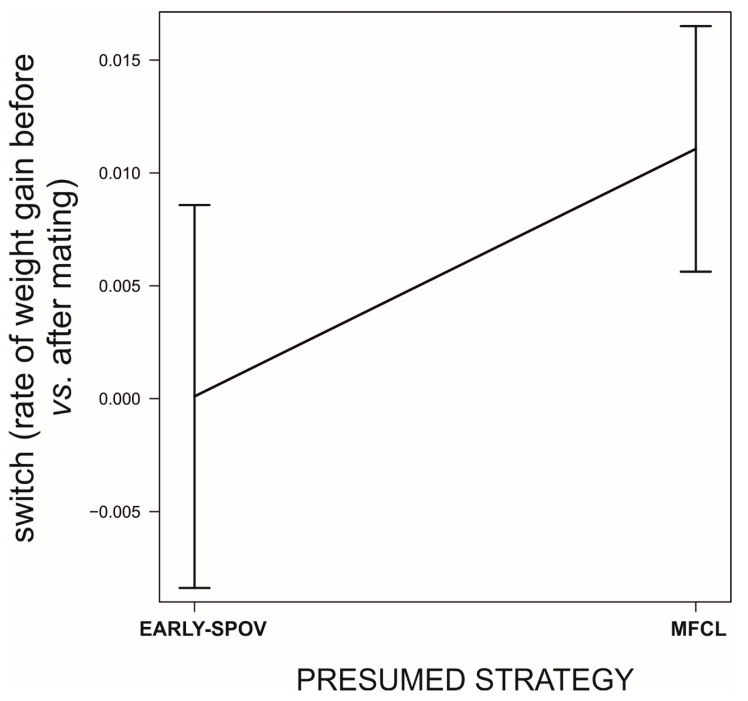
Differences in switching in the rate of food acquisition (difference before vs. after first mating) between females that killed a male before the female had first mated (presumed EARLY-SPOV) and females that killed their first male after the female first mated (presumed MFCL). Data source as in Figure 5.

**Figure 9 biology-09-00241-f009:**
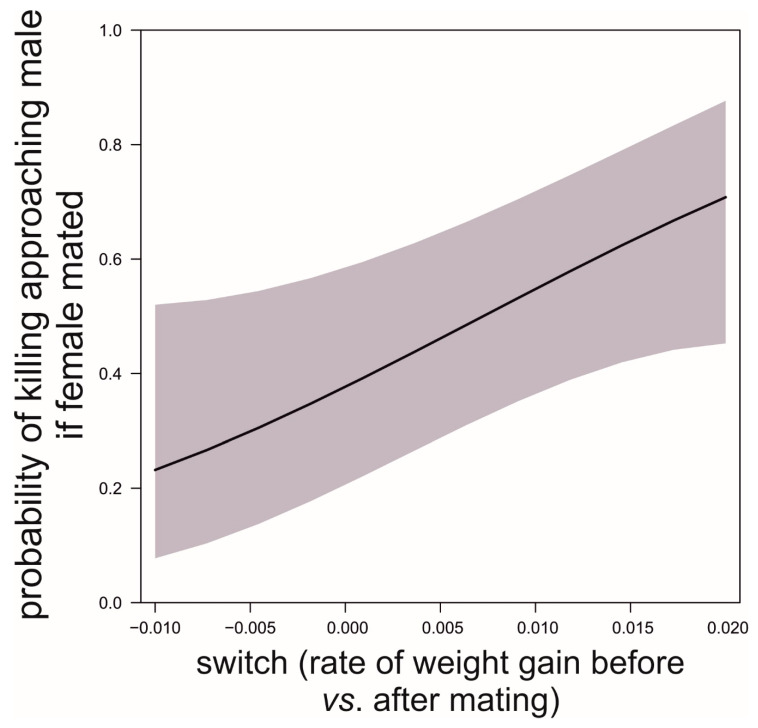
Probability of females killing males that approach females once the latter have mated as a function of switching in the rate of food acquisition (difference before vs. after first mating). Data source as in Figure 5.

**Figure 10 biology-09-00241-f010:**
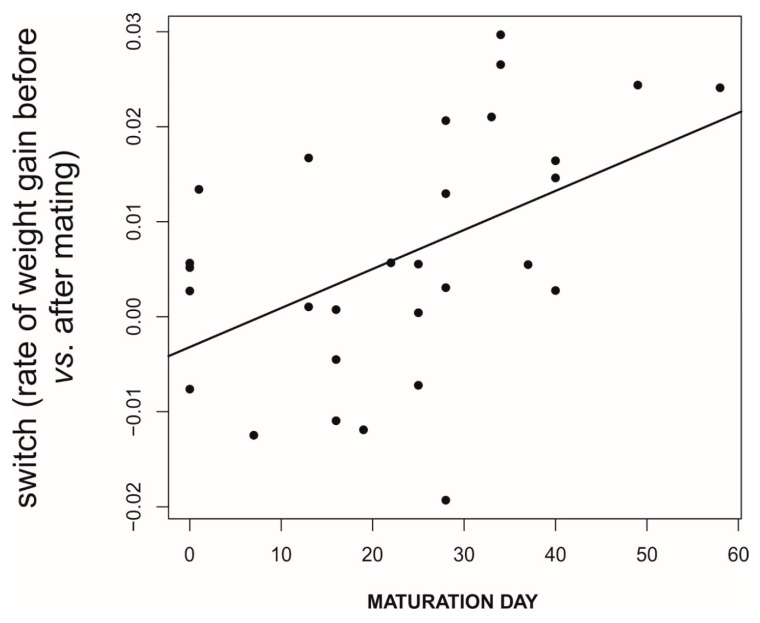
Females that mature later have a higher switching rate of food acquisition (difference before vs. after first mating), as expected if they followed the MFCL strategy. Data source as in Figure 5.

**Table 1 biology-09-00241-t001:** List of parameters as calculated from studies in *Lycosa hispanica* (sources: [34,39,49,63,64] J. Moya-Laraño *unpublished*).

Parameter	Value/s
Date of Season’s Onset	April 23rd
Season length	79 days
Background female mortality rate for MFCL	0.0030 day^−1^
Background female mortality rate for SPOV, BCD-SPOV, EM-SPOV and BCD EM-SPOV *	0.0036 day^−1^
Background male mortality rate	0.0045 day^−1^
Female maturation time, “fem_mat” (since April 23rd) for SPOV, BCD-SPOV and MFCL	N|μ = 56, σ = 0.5|
Female maturation time, “fem_mat” (since April 23rd) for EM-SPOV and BCD EM-SPOV	N|μ = 39, σ = 0.5|
Male maturation time, “mal_mat” (since April 23rd)	N|μ = 36, σ = 0.5|
Adult body size, CW (mm) for EM-SPOV and BCD EM-SPOV	N|μ = 3.21 + 0.047 * fem_mat + 0.39 * CL, σ = 0.5066|
Adult body size, CW (mm) for MFCL, SPOV and BCD-SPOV	N|μ = 3.21 + 0.024 * fem_mat + 0.39 * CL, σ = 0.4679|
Initial condition, COND_o_ (mm)	N|μ = 3.54 + 0.49 * CW, σ = 0.2671|
Threshold level for female satiation (maxCOND_f_, mm) **	−38.98 + 11.73 * CW − 0.63 * CW^2
Added value on condition from feeding on a male	2.39 mm (~0.199 g)
Egg sac volume, “vol” (mm^3^) for SPOV and EM-SPOV	N|μ = −1156.43 + 277.21 * CW, σ = 140.33
Egg sac volume, “vol” (mm^3^) for BCD SPOV and BCD EM-SPOV	N|μ = −1156.43 + 277.21 * CW + 123.44 * maxCOND_f_, σ = 132.97| ****
Egg sac volume, “vol” (mm^3^) for MFCL	N|μ = −2297.64 + 217.88 * CW + 123.44 * maxCOND_f_, σ = 129.59| ****
Offspring number (“N”)	N = 57.54 + 0.16 * vol
Daily increase in condition (or daily net abdomen growth mm/day) for MFCL	U|0,1| * 0.1482 ***
Daily increase in condition (or daily net abdomen growth in mm/day) for SPOV, BCD-SPOV, EM-SPOV and BCD EM-SPOV	1.5 * U|0,1| * 0.1482 ***
Maximum daily rate of encounter with females, “maxenc”	1, 3 day^−1^
Probability of an SPOV female attacking a male, “pspov”	0.5, 0.9
Probability of male escaping from a female attack, “pescape”	exp(−CW * 0.1)
Proportion of offspring surviving to maturation in SPOV, BCD-SPOV, EM-SPOV and BCD EM-SPOV relative to MFCL, “different”	0.1, 0.3, 0.5, 0.7, 0.9

* Because higher foraging effort is required in poor environments 0.0036 is replaced by 0.0045 (i.e., the parameter “mort” takes values 1.2 or 1.5, see Appendix A). ** Females with condition (mm) above this threshold do not feed anymore and are ready to lay an egg sac. Satiated females will attack males only if SPOV. *** This reflects prey availability in addition of voracity. In poor environments 0.1482 is replaced by 0.0741. **** This value of σ is for maximum condition, in reality it is a state-dependent variable.

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
