# Peer review of "Model and Data Concur and Explain the Coexistence of Two Very Distinct Animal Behavioral Types"

_biology, 2020, doi:10.3390/biology9090241_

Round 1
Reviewer 1 Report
This manuscript develops a novel model exploring cost and benefits for sexual cannibalism in spiders and how different behavioural patterns can invade and persist in a population. It is unique in using real data of a population of Lycosa hispanica for testing and re-evaluating the model assumptions.
Introduction, methods results and discussion are very well written, generally concise and well-focused on the topic at hand. However, the authors should probably try to find some ways to shorten the manuscript. For example, there is very little value in the first sentence of the introduction!
I really enjoyed reading this manuscript and found very little that requires major attention, although to appeal to the more general audience it would be interesting to know if there are similar studies in other invertebrates or vertebrates. The authors mainly refer to a fishing spider study in their discussion.
Minor issues: p. 3, line 110 (and other places in manuscript): the term "formally L. tarantula" is somewhat misleading. A species can't have been formerly another species. Rephrase to indicate it was previously misidentified.
p. 8, l.94: should be "less, not more..." (instead of "no")?
Author Response
This manuscript develops a novel model exploring cost and benefits for sexual cannibalism in spiders and how different behavioural patterns can invade and persist in a population. It is unique in using real data of a population of Lycosa hispanica for testing and re-evaluating the model assumptions.
Introduction, methods results and discussion are very well written, generally concise and well-focused on the topic at hand. However, the authors should probably try to find some ways to shorten the manuscript. For example, there is very little value in the first sentence of the introduction!
R: Thanks so much for your comments. Since we do not surpass the word limit, we include both simulation and data, and had only 5 days to respond, we have decided not to shorten the ms. We removed part of that sentence you referred to though (Now Line 43).
I really enjoyed reading this manuscript and found very little that requires major attention, although to appeal to the more general audience it would be interesting to know if there are similar studies in other invertebrates or vertebrates. The authors mainly refer to a fishing spider study in their discussion.
R: Unfortunately the answer is no. However, we have made an effort to appeal to a wider audience by including a new introductory paragraph (Lines 31-42) and sentences at the end of the Intro (Lines 124-129), also in response to reviewer 3.
Minor issues: p. 3, line 110 (and other places in manuscript): the term "formally L. tarantula" is somewhat misleading. A species can't have been formerly another species. Rephrase to indicate it was previously misidentified.
R: Well, not exactly. Through phylogeny experts change the identity of species (even families and genera) quite often. We have now included the authorities of the species names and the article where the species-population is transferred from one species to another. Lines 123-124.
p. 8, l.94: should be "less, not more..." (instead of "no")?
R: yes, thanks, changed in P9 L99
Reviewer 2 Report
General comment
Using both modelling and field data, this study brings some interesting and original conclusions on how behavioral syndromes can co-exist in animal populations. This ms is very well written, data analysis and modelling sound, and I only have some minor suggestions that can hopefully help improving a bit the ms.
Minor comments
Title: Maybe add how model and data act for explaining the coexistence of behaviors (e.g. both, jointly, independently, etc.).
L24: precise “using field data”?
L26 (and L28): Not fully sure how directly relates to animal personalities, which are not properly introduced and discussed further. Maybe stick to the coexistence of
L32: “individual behavior” -> “individual behaviors”, no?
L36-58: For any raisons, I find this part a bit hard to follow, probably too broad and not specific enough.
L64: Is this an acronym from your own? What does it mean?
L64: Would be nice to know how widespread is this syndrome in animal tree, spiders being “just an example”.
L75 (and not L87): I’d say that you’re still on spiders.
L106: Maybe provide the software name later? Is it available online?
L110: Move the synonymy and provide taxonomic authorities on L160.
L125-126: Is this a realistic assumption (egg number and volume were repeatedly found negatively correlated in lycosids)?
[Table A1 of Electronic Appendix was not available]
L13 (of P7): could be worth citing Jakob et al. 1996 // Oikos.
L67: please provide the year of fieldwork.
L129-140, 207-210: Actually Methods, but makes sense to leave these there.
L147, 178-180, 187-189, 197-205: more Discussion than Results s. st. (some parts could feed the discussion, sections like 4.1. are short)
L226-230: At this stage, I was wondering if the ms would not read better if assumptions were placed at the introduction (which currently ends very general)? See also e.g. L.111 and L 205-207.
L265: should actually write ‘à la carte’!
L259-329: very convincing text!
L363-365: Need citation(s)
L521: Journal name is provided two times.
L584: This is a PhD thesis, right?
Author Response
General comment
Using both modelling and field data, this study brings some interesting and original conclusions on how behavioral syndromes can co-exist in animal populations. This ms is very well written, data analysis and modelling sound, and I only have some minor suggestions that can hopefully help improving a bit the ms.
Minor comments
Title: Maybe add how model and data act for explaining the coexistence of behaviors (e.g. both, jointly, independently, etc.).
R: Great idea, we have now entitled the ms: “Model and data concur and explain the coexistence of two very distinct animal behavioral types”
L24: precise “using field data”?
R: It would be nice but there is no sufficient space in the 200-word limit to be more precise.
L26 (and L28): Not fully sure how directly relates to animal personalities, which are not properly introduced and discussed further. Maybe stick to the coexistence of
R: the two very distinct behavioral types can only reflect differences in personalities. Even though we do not make this explicit, we believe that this being a special issue on animal personalities, it is appropriate to end the Abstract pointing to that fact.
L32: “individual behavior” -> “individual behaviors”, no?
R: yes, correct. We changed it accordingly..
L36-58: For any raisons, I find this part a bit hard to follow, probably too broad and not specific enough.
R: We indeed tried to appeal at a broad audience, and we become narrower later in the Intro. Indeed, Reviewer 3 asked to make the Intro even broader and we introduced a paragraph to put our paper in an even broader contest.
L64: Is this an acronym from your own? What does it mean?
R: Exactly: SPillOVer, we have made this clearer on L76.
L64: Would be nice to know how widespread is this syndrome in animal tree, spiders being “just an example”.
R: We only know to affect spiders. Sexual cannibalism is not that frequent across the animal kingdom.
L75 (and not L87): I’d say that you’re still on spiders.
R: We have now introduced the former L87 with “Across taxa,” to then continue talking about spiders (now L100).
L106: Maybe provide the software name later? Is it available online?
R: L106 refers to the additional analyses provided as supplementary material, not to the software, which is indeed available on-line with a link on L 164.
L110: Move the synonymy and provide taxonomic authorities on L160.
R: Lines do not seem to match anymore your version and that reformatted by Biology. In any case, we provided the authorities and the citation for the transfer on L 123-124.
L125-126: Is this a realistic assumption (egg number and volume were repeatedly found negatively correlated in lycosids)?
R: Yes, you are absolutely right and found that relationship in this species (unpublished data), however this would involve complicating the model unnecessarily (i.e. it would barely change the conclusions). Besides, we do not have a good function relating offspring body mass to survival in the wild for cursorial spiders. See Rabaneda-Bueno et al. 2008 (PLoSOne) were neither body size nor body condition affected offspring survival in field enclosures for this species. We have now started the assumption with “For simplicity…”. Line 144.
[Table A1 of Electronic Appendix was not available]
R: Sorry about that. An Excel file was uploaded with the submission. We hope this will become available to the readership.
L13 (of P7): could be worth citing Jakob et al. 1996 // Oikos.
R: certainly this reference on body condition is not relevant for that line in our version, which states: “Female body size (i.e. carapace width, CW) was determined by its observed relationship to maturation time.”. We have cited it here: “A feeding algorithm allowed each female spider to increase her condition (abdomen width, Jakob et al. 1996) each day by a factor dependent on the strategy and on the environment (see Environments below), simulating foraging encounters.” where we believe is most appropriate. Now line 161.
L67: please provide the year of fieldwork.
R: done on now P9 L68.
L129-140, 207-210: Actually Methods, but makes sense to leave these there.
R: Ok, agreed.
L147, 178-180, 187-189, 197-205: more Discussion than Results s. st. (some parts could feed the discussion, sections like 4.1. are short)
R: As lines do not match anymore we believe that you mean interpretations of results on this section: “Field data: do the simulated strategies coexist in the wild?” These interpretations of results are necessary to link one test to the next and one prediction to the next test. They make this empirical section to flow better. In any case 4.1 speaks only about simulated results, not empirical.
L226-230: At this stage, I was wondering if the ms would not read better if assumptions were placed at the introduction (which currently ends very general)? See also e.g. L.111 and L 205-207.
R: The assumptions belong to the modeling section only. The Intro belongs to the entire ms. We have however expanded the end of the Intro to make it more appealing to a broader readership.
L265: should actually write ‘à la carte’!
R: Thanks, yes, in French! Now changed.
L259-329: very convincing text!
R: Thanks so much!
L363-365: Need citation(s)
R: We added the citations: Gould 1987; Newman and Elgar 1991.
L521: Journal name is provided two times.
R: Thanks, corrected.
L584: This is a PhD thesis, right?
R: yes, corrected. Thanks.
Reviewer 3 Report
Introduction starts and stays too narrow for me. I think a broad readership will enjoy this paper, but as its written now it does not give a sufficient intro to maximize that readership.
We need a better picture of the hyp and predictions by the end of the Intro. This is especially true given you have theoretical and empirical results.
If Im not mistaken Table 1 contains abbreviations that have not yet been introduced.
I have no expertise to comment on the modelling methodology, as I am an avowed experimentalist. That said, I was genuinely compelled by this work. I agree with the Discussion's first paragraph that this is a great example of weaving back and forth from theory to data and back again. The spillover hypothesis has been roughly treated by many in the literature. I admire this work's efforts to look for contingencies where spillover can survive. I would suggest that the modified early spillover strategy is a perfectly likely situation and the authors should make sure to review the spider literature that supports this notion of payoffs for high levels of voracity in juveniles. Especially given how rare a MFCL strategy appears to me in my experience.
I have very few qualms about this work. I think it is quite well done.
Author Response
Introduction starts and stays too narrow for me. I think a broad readership will enjoy this paper, but as its written now it does not give a sufficient intro to maximize that readership.
R: We have now included an introductory paragraph (L31-42)
We need a better picture of the hyp and predictions by the end of the Intro. This is especially true given you have theoretical and empirical results.
R: Some of the predictions are made by the model and cannot be made so early in ythe ms. However, we made an effort to appeal to a wider audience including some predictions and talking about the big picture of this work (L 124-129).
If Im not mistaken Table 1 contains abbreviations that have not yet been introduced.
R: Yes, they were included and introduced there intentionally as not to interrupt the flow of the paper.
I have no expertise to comment on the modelling methodology, as I am an avowed experimentalist. That said, I was genuinely compelled by this work. I agree with the Discussion's first paragraph that this is a great example of weaving back and forth from theory to data and back again. The spillover hypothesis has been roughly treated by many in the literature. I admire this work's efforts to look for contingencies where spillover can survive. I would suggest that the modified early spillover strategy is a perfectly likely situation and the authors should make sure to review the spider literature that supports this notion of payoffs for high levels of voracity in juveniles. Especially given how rare a MFCL strategy appears to me in my experience.
R: Thanks so much. Ironically, the first author was attracted by the ASH because in his experience (field and laboratory) wolf spider females waited to mate to have sperm. SPOV seems not to be very prevalent in our populations. We did find only 8 cases of virgin females killing males. We indeed need to know what are the payoffs for juveniles in the wild. This, however, has not been investigated much because researchers tend to work with adults (but perhaps work by Susan Riechert with Agelenopsis aperta which is not a sexually cannibalistic species). Makes sense juveniles lost marking tags through molting and are difficult to track.
I have very few qualms about this work. I think it is quite well done.
R: thanks very much.